# Nutritional Assessment of Plant-Based Meat Alternatives: A Comparison of Nutritional Information of Plant-Based Meat Alternatives in Spanish Supermarkets

**DOI:** 10.3390/nu15061325

**Published:** 2023-03-08

**Authors:** Lucía Rizzolo-Brime, Alicia Orta-Ramirez, Yael Puyol Martin, Paula Jakszyn

**Affiliations:** 1Unit of Nutrition and Cancer, Cancer Epidemiology Research Programme, Catalan Institute of Oncology (ICO), Bellvitge Biomedical Research Institute (IDIBELL), L’Hospitalet de Llobregat, 08908 Barcelona, Spain; 2Blanquerna School of Health Sciences, Ramon Llull University, 08025 Barcelona, Spain

**Keywords:** plant-based meat alternatives, protein alternatives, meat analogues, nutritional quality, sustainability

## Abstract

Since the classification of processed meat as carcinogenic by the International Agency for Research on Cancer (IARC) in 2015, an increase in consumption of plant-based meat alternatives (PBMAs) has been observed worldwide. This occurs in a context characterized by concern for health, animal welfare, and sustainability; however, evidence of their nutritional quality is still limited. Therefore, our objective was to evaluate the nutritional profile and processing degree of PBMAs available in Spain. In 2020, products from seven Spanish supermarkets were analyzed for their nutritional content and ingredients. Of the 148 products, the majority were low in sugars but moderate in carbohydrates, total and saturated fat, and high in salt. The main vegetable protein sources were soy (91/148) and wheat gluten (42/148). Comparatively, 43/148 contained animal protein, the most common being egg. Overall, PBMAs had a long list of ingredients and additives, and they were classified as ultra-processed foods (UPFs) according to the NOVA system. This study shows that the PBMAs available in Spanish supermarkets have a variable nutritional composition within and between categories. Further research is needed to determine if replacing meat with these UPFs could be a good alternative towards healthier and more sustainable dietary patterns.

## 1. Introduction

In October 2015, the International Agency for Research on Cancer (IARC) released a report that classified unprocessed red meat as “probably carcinogenic” and processed meat as “carcinogenic” to humans based on the association between consumption of red and processed meat and colorectal cancer [1]. In this context, there has been an increasing number of people removing meat from their diets, or at least reducing its consumption, and asking for alternative products [2,3,4]. Consequently, there has been a considerable need to increase the offer of plant-based meat alternative products (PBMAs) to please consumers. In fact, plant-based products sales increased 49% from 2018 to 2020 in Europe [5] and, in Spain alone, the sales volume of PBMAs increased by 32% from 2019 to 2020 [6]. This trend has been observed worldwide as reported in studies done in the UK, Brazil, Norway, the US, Australia and Germany [7,8,9,10,11,12]. Among the reasons for the growing consumption of PBMAs, however, in addition to the health issues, is also concern for animal welfare, greenhouse gas emissions, and overall environmental impacts attributed to meat production [13]. Curiosity and interest in trying new products has also been identified as a fundamental reason for why consumers are motivated to try plant-based products [14].

Currently, despite the nutritional information on PBMAs, there is need for a regulatory framework to allow the population to have better food choices [15]. Furthermore, manufacturers frequently use nutrition and health claims on food labels to emphasize the positive nutritional attributes of their products, which can influence consumer purchasing and food choices [16]. Additionally, although there is some evidence that PBMAs may contain similar amounts of calories, protein and iron as the meats they are supposed to replace [17] and some studies show that PBMAs may have a relatively better nutritional composition [18], there is not sufficient evidence to consider them healthier than their meat counterparts. In fact, most of the available PBMAs are manufactured in the form of sausages, burgers or nuggets and contain higher amounts of sodium, oil and additives, including coloring, flavoring and binding agents, in comparison to unprocessed meats [19,20]. Several studies have shown that the avoidance of animal-based food appears to be associated with the introduction of ultra-processed PBMAs over more natural plant sources, compromising the general quality of a plant-based diet [21,22,23]. Moreover, ultra-processed food has been associated with several health problems, like overweight and obesity, as well as metabolic syndrome prevalence, LDL cholesterol, risk of hypertension, and higher risks of cardiovascular, coronary heart, and cerebrovascular diseases [24,25].

NOVA is a food classification system based on the degree of processing. In general, the classification criteria are based on the nature, extent, and purpose of industrial processing [26,27].

Thus, our aim was to evaluate the nutritional composition and processing degree and to perform a quantitative analysis of the ingredients of PBMA products currently available in Spanish supermarkets.

## 2. Materials and Methods

### 2.1. Product Description and Classification

Between September 2020 and December 2021, a sampling of PBMAs was conducted in seven of the most common supermarket chains in Spain, namely Alcampo, Aldi, Carrefour, Dia, El Corte Inglés, Lidl and Mercadona. These chains are well positioned throughout the country and were chosen to represent choices available to the majority of the Spanish population [28]. For each product, the following data were collected: name, list of ingredients, nutritional content, and organic or conventional farming. In terms of nutritional content, the following information was collected: calories, carbohydrates, sugars, proteins, total fat, saturated fat, salt, and fiber per 100 g of product. This information was obtained from the nutritional label of the product packages (through photographs) and double-checked from the website where the product was published.

Spanish PBMAs were categorized into eight different food groups according to their format characteristics: (1) plant-based sausages, (2) plant-based nuggets and breaded food, (3) plant-based meatballs, (4) vegetarian cold cuts, (5) veggie patties, (6) “Beyond Meat-type” hamburger patties, (7) “Chicken-type” strips (vegan/vegetarian products made from soy), and (8) plant-based mince. Each item was classified into a single main group, based on the similarity to meat-based products and dishes. Products excluded from the study were those vegetarian foods not specifically created to mimic meat products, such as tofu, tempeh, seitan and falafel.

In order to classify the products by their nutritional composition, we used nutritional claims from AESAN [29] (Table 1). However, in some cases, the UK’s FoP labeling [30] was used (according to the nutrition claims of EU Nutrition and Health Claims Regulation legislation (EC) 1924/2006) [31] in order to categorize the products based on the content of total fat, saturated fat and salt as shown in Table 2.

To classify the products according to the degree of processing, we used the NOVA system [26,27] and the four groups currently described:

1. Unprocessed or minimally processed foods; 2. Foods with a simple or basic processing; 3. Moderately processed foods and 4. Ultra-processed foods.

### 2.2. Statistical Analysis

Descriptive statistics (median, minimum and maximum) of energy and selected nutrients (carbohydrates, sugars, fat, saturated fat, protein, sodium and dietary fiber) were calculated per 100 g. As expected, there was missing information about dietary fiber and iron, unless specifically added to these products. Groups were assigned according to their main ingredients and similar characteristics to comparable processed meat products. The statistical analysis was performed using the statistical software Rstudio Desktop version 4.1.2, with the significance level set at *p* < 0.05.

## 3. Results

Overall, 176 plant-based meat alternatives were identified, but due to the repetition of some products in the different Spanish supermarkets, we analyzed a total of 148 individual products. Alcampo supermarket had the largest number of products on offer (*n* = 54), whereas the other supermarkets had lower quantities of products available (Aldi *n* = 28, Carrefour *n* = 30, El Corte Inglés *n* = 24, Dia *n* = 5, Lidl *n* = 26 and Mercadona *n* = 11). Table 3 shows the plant-based product categories analyzed in this study. The products were classified into eight different categories according to their similarity to comparable processed meats. Veggie patties were the group most found in supermarkets (29/148), followed by plant-based meatballs (25/148), plant-based sausages (21/148), “Beyond Meat-type” hamburger patties (20/148), vegetarian cold cuts (17/148), plant-based mince (17/148), “Chicken-type” strips (10/148) and plant-based nuggets and breaded food (9/148). We evaluated every group in order to compare quantitative and qualitative information and estimate the degree of processing using the NOVA classification system.

### 3.1. Ingredient Analysis

Table 4 summarizes the main ingredients of the products analyzed. The main protein sources of these products diverged greatly depending on the item. The most common was soy protein (91/148), used in a variety of forms, including concentrated soy protein, isolated soy protein, hydrolyzed soy protein and texturized soy protein. The second most common was gluten (42/148). A variety of whole grains and flours such as black beans, chickpeas, rice, spelt, corn and quinoa were present in 64/148 of the products. However, 43/148 of the products contained some type of animal protein, making them non-vegan products. The main animal protein sources were egg white and milk. Overall, vegetarian cold cuts were the category that had more animal protein content in their ingredients, followed by plant-based nuggets and breaded food, and plant-based meatballs.

With regard to fat, only a few products reported the percentage of the main oil that they contain. The most common was sunflower oil, present in 101/148 of the products, followed by rapeseed oil, reported in 44/148 of the items. The use of saturated fats like coconut oil or palm oil was significantly lower (15/148 and 2/148, respectively). Only 8/148 of the products contained olive oil exclusively in their formulation and 5/148 of the products had a combination of olive oil and sunflower or rapeseed oil. Egg yolk was present in 9/148 products, making them non-vegan. Considering carbohydrate sources, wheat flour was the most common ingredient, present in 64/148 of the products. Only 11/148 of the products were fortified with iron and 10/148 with vitamin B12. Overall, only 21/148 of the products were fortified.

### 3.2. Organic Farming

There was a diversity of the percentages of the products labeled as organic (Table 5) in the different categories, but the majority of the products available in the supermarkets were produced conventionally. Organic vegetable protein was found in 52/148 of the products, whereas 45/148 contained organic fat and only 3/148 of products contained organic animal protein.

### 3.3. Nutritional Composition

Table 6 shows the nutritional information by group as the median (minimum and maximum) per 100 g of product across each category. The average energy was 215 kcal/100 g. Plant-based mince had the highest value (320 kcal/100 g) followed by plant-based sausage (254 kcal/100 g), plant-based nuggets and breaded food (229 kcal/100 g), and veggie patties (201 kcal/100 g), whereas “Chicken-type” strips had the lowest energy (154 kcal/100 g). Regarding the protein content, the average was 15.0 g/100 g, with vegetarian cold cuts showing the lowest content (7.30 g/100 g) and plant-based mince the highest (47.30 g/100 g).

Total carbohydrates ranged from 2.65 g to 17.9 g/100 g. Plant-based nuggets and breaded food (17.9 g/100 g), veggie patties (15.6 g/100 g) and plant-based mince (13.0 g/100 g) contained the highest amounts. Sugars were generally low (<5 g/100 g) and only plant-based mince had a moderate content (5.80 g/100 g). Almost all products can be considered a source of fiber (>3 g/100 g).

In regard to total fat and saturated fat, the average values, 10.0 g/100 g and 1.30 g/100 g, respectively, can be considered moderate. Most products were upper-moderate to high in salt (>1 g/100 g), with vegetarian cold cuts having the highest content (2.00 g/100 g) and excepting the plant-based mince group, which presented the lowest (0.07 g/100 g) [30].

According to the NOVA classification system, 93.9% of the products were categorized as ultra-processed food (Group 4) and the remaining 6.08% of PBMAs were categorized as processed food (Group 3, for the plant-based mince group).

## 4. Discussion

According to recent studies, PBMA consumption is growing worldwide, reflecting a shift to a diet rich in vegetables and low in animal protein [32,33]. The recent report of the WHO’s International Agency for Research on Cancer (IARC) in which processed meat has been classified as carcinogenic for humans (Group 1) and red meat as a probable cause of cancer (Group 2A) in humans [1] has caused a great concern within the scientific community as well as for consumers. Alternative strategies for change include reducing the size of meat pieces or increasing the consumption of vegetable proteins [32]. As a consequence, there is a new market niche that includes PBMAs, and the impact of these products on customers’ daily food decisions has increased.

Although more studies assessing the nutritional profile and healthiness of PBMAs are being published [18,34,35], to our knowledge, the present study is the first that not only describes the nutritional composition but also the processing degree of PMBAs available in Spanish supermarkets and, in addition, is the first to provide a quantitative analysis of the ingredients of these PBMAs. Our findings indicate that PBMAs are quite widespread in Spanish supermarkets (176 products available in seven supermarkets analyzed). This is in agreement with the worldwide trend of increased consumption of PBMAs and with the fact that this type of products is in demand, accepted and used not only by vegetarians and vegan consumers, but is also included in omnivore diets [36,37]. However, although PBMA consumers demand these products for reasons like health, animal welfare, the environment and rejection of meat, there is still not enough evidence to support that all plant-based products are as healthy and/or sustainable as they appear to be. As seen in our results, most PBMAs contained high salt and the vast majority were classified as UPFs according to the NOVA system. A study by Hu et al. [38] found that replacing meat with a plant-based substitute does not necessarily reflect a healthy dietary pattern. Khandurpur et al. [39] reported that within vegetarian diets, vegans were the largest consumers of UPFs, followed by lacto-ovo vegetarians and pescatarians. Recent studies have shown that consumption of UPFs is associated with greater caloric intake, weight gain and, thus, negative health outcomes [40,41].

Our results agree with other studies which examined the nutritional information of PBMAs, reporting similar energy values [8,12]. Similarly, according to the British Nutrition Foundation (BNF), PBMAs had medium energy density (1.5–4 kcal/g) [42]. Taking into account that weight gain is associated with excessive energy intake [43], energy derived from consumption of PBMAs should be considered by policy makers when implementing actions that promote consumption of fresh or minimally processed plant-based products in order to prevent secondary diseases [44].

The total fat and saturated fat in PBMAs were moderate and varied substantially in all categories. The most common fat sources for the majority of products were sunflower oil and rapeseed oil. It has been shown that sunflower oil induced an increase of different proinflammatory markers [45] and that intake of polyunsaturated fatty acids (PUFA-fatty acid identified in sunflower oil) was highest in vegans, followed by pescatarians, semi-vegetarians and vegetarians, and lowest in meat-eaters [46]. Even though the PBMAs showed a moderate fat content, overconsumption can lead to a diet high in fat which can be associated with unfavorable changes in gut microbiota, fecal metabolic profiles and plasma proinflammatory factors, which could have unfavorable consequences for long-term health outcomes [47]. Our results agreed again with other studies; for example, in the plant-based sausages category, which shows similar total fat ranges [8,11,12] as well as common fat sources [11,48].

Regarding proteins, the main protein sources in Spanish PBMAs were soy and gluten, which agrees with other studies done on products found in Germany or Brazil [12,49]. Unfortunately, these two vegetable proteins are considered common allergens [50,51]. According to the SEOM (Sociedad Española de Oncología Médica), plant-based nuggets and breaded and veggie patties had a medium content of proteins (10–15 g/100 g), whereas “Beyond Meat-type” hamburger patties, plant-based meatballs, plant-based sausages, “Chicken-style” strips and plant-based mince had a high content (>15 g/100 g) [52]. On the other hand, it has been shown that plant protein, as a part of plant-based diet, is associated with improvements in body composition and reduction in both body weight and insulin resistance [53]. Interestingly, many products contained combinations of different protein sources, including animal protein. This fact is important in order to make a good, safe and true nutritional claim for the vegan population. In this context, leghemoglobin (LegH) protein from soy, an analogue of myoglobin, performs a crucial role: when it is cooked, it unfolds, releasing, similarly to myoglobin, the heme cofactor to catalyze reactions that result in the variety of compounds that define the exceptional aroma and flavor of meat [54]. Moreover, the heme cofactor of LegH is identical to the heme found in animal meat [55]. Some studies have evaluated the safety of LegH, establishing a no-observed-adverse-effect level of 750 mg/kg/d LegH, which is over 100 times greater than the 90th percentile estimated daily intake [54]. On the other hand, one third of the products had methylcellulose (E461) added to their ingredients. It has been shown that after an overdose of methylcellulose, or for people who are allergic to it, problems like hives, breathing difficulty, and/or swelling of the face, lips, tongue or throat can occur [56].

PBMAs showed variable content in carbohydrates and were low in sugars due to the presence of pulses in their ingredients. Pointke et al. [12] reported a similar overall carbohydrates and sugars content. Our results also agreed with Bryngelsson et al. [48] in the plant-based nuggets and breaded foods category. PBMAs may be responsible for contributing a part of the carbohydrates and sugars in diets, as they are added as flours or starches as well as gums.

Salt content was substantially high in most PBMAs. The WHO recommends consuming less than 5 g/day [57]. Considering the categorization of foods by their salt content, only the category of plant-based mince had a low salt content, and the category of vegetarian cold cuts had a high content of salt. Different studies have shown similar results in relation to high salt content in PBMAs [11,58]. At the end of 2020 in Spain, consumption of salt peaked at 14.8% [59]. It has been observed that about 75% of dietary salt comes from consuming processed foods, so it is an important aspect to consider for consumers’ nutrition guidelines [60]. As PBMAs continue to grow in their number of product categories as well as consumer acceptance, it is important to highlight the need for nutrition guidelines in their development.

In addition to comparing the nutritional facts of PBMAs, they were also categorized using the NOVA classification. Results have shown that almost 94% of the products were classified as UPFs (4. Ultra-processed foods). The NOVA classification is one of the most referenced in the literature and it has been applied in several countries [61]; however, it has been notably criticized for problems with the correct definition of ultra-processed food, for example: it does not define critical nutrients, it does not allow micronutrient intake to be quantified, and some examples do not match with the classification, among others. Nevertheless, PBMAs are characterized as an ultra-processed food [62], a type of products which is associated with noncommunicable diseases or their risk factors [63,64,65]. In general, UPFs are high in unhealthy types of fat, free sugars and salt and refined starches, are energy-dense, and are poor sources of dietary fiber, protein and micronutrients [66]. For these reasons, it is important to provide consumers with adequate information that reflects not only the nutritional quality of the products, but the processing level too.

The present study is the first that describes the nutritional composition and processing degree of PMBAs available in Spanish supermarkets. In addition, it provides a quantitative analysis of the ingredients of Spanish PBMAs, allowing the comparison of the nutritional quality between and within the eight categories. However, some limitations should be mentioned. First of all, we did not include all supermarket chains in the study, nor all products that are exclusively found in vegan specialty stores. Further research is needed to identify some nutrition gaps, such as the micronutrient content and bioavailability, protein quality and digestibility, and dietary fiber type found in PBMAs.

## 5. Conclusions

This study shows that a great number of plant-based meat alternative products available in Spanish supermarkets have a variable nutritional composition depending on the product category. There exists a false belief about the healthiness of these products because of their plant origin. Although PBMAs may show medium to high contents of vegetable protein and can be considered sources of dietary fiber, the majority of these products also meet the criteria of ultra-processed food, so their consumption should be sporadic within a plant-based diet with fresh vegetables, fruit and legumes. After a closer examination, both nutritional and quality information about PBMAs is needed to develop guidelines to counsel consumers about including these products in a healthy plant-based diet.

To our awareness, the present study was the first one to focus on plant-based meat alternatives available in Spanish supermarkets, analyze the nutritional information, and estimate NOVA classifications for these products. Further research is needed to determine if replacing processed or unprocessed foods with plant-based meat alternatives in people’s diets can eventually lead to healthier dietary patterns.

## Figures and Tables

**Table 1 nutrients-15-01325-t001:** Nutrition claims authorized in the annex of Regulation (EC) Nº 1924/2006.

Health Claim	Use Condition
Low energy value	<than 40 kcal (170 kJ)/100 g (solids) or >than 20 kcal (80 kJ)/100 mL (liquids).
Low fat content	<than 3 g/100 g (solids) or 1.5 g/100 mL (liquids), 1.8 g/100 mL for semi-skimmed milk.
Low saturated fat content	The sum of saturated and trans-fatty acids < 1.5 g/100 g (solids) and <0.75 g/100 mL (liquids), and in any case the sum of saturated fatty acids and trans-fatty acids should not provide more than 10% of the energy value.
Low sugar content	<5 g/100 g (solids) or 2.5 g/100 mL (liquids).
Low sodium/salt content	<0.12 g of sodium, or the equivalent value of salt, per 100 g or per 100 mL.
Source of fiber	≥3 g of fiber per 100 g or ≥1.5 g of fiber per 100 kcal.

Adapted from the Agencia Española de Seguridad Alimentaria y Nutrición, Ministerio de Sanidad, Consumo y Bienestar Social, Tabla de declaraciones nutricionales, 2019, pp. 1–5 [29].

**Table 2 nutrients-15-01325-t002:** Classification of critical nutrients.

Levels	Total Fat (g/100 g)	Saturated Fat (g/100 g)	Salt (g/100 g)
Low	<3	≤1.5	≤0.3
Medium	>3 to 17.5	1.5 to 5	0.3 to 1.25
High	>17.5	>5	>1.25

Data from UK’s FoP labeling (according to the nutrition claims of EU Nutrition and Health Claims Regulation legislation (EC) 1924/2006) [31].

**Table 3 nutrients-15-01325-t003:** Classification of plant-based product categories.

Category	Total Products (*n* = 148)	Description
Plant-based sausages	21	Feature either “sausage” or “hot dog” in the product name.
Plant-based meatballs	25	Small, meat-free balls appearing to mimic beef before cooking.
Plant-based nuggets and breaded food	9	Bite-size meat-free pieces appearing to mimic chicken, usually batter-fried.
Vegetarian cold cuts	17	Feature “cooked meats slices” in the product name.
Veggie patties	29	Meat-free patties, including either “burger”, and/or “pattie/patty” in the product name.
“Beyond Meat-type” hamburger patties	20	Plant-based meat substitute that looks and tastes like real ground beef, including either “burger”, and/or “pattie/patty” in the product name.
“Chicken-type” strips	10	Meat-free strip or stick, appearing to mimic chicken.
Plant-based mince	17	Feature “mince” in the product name.

**Table 4 nutrients-15-01325-t004:** Ingredients in plant-based meat alternatives contributing to key nutrients.

Nutrient	Ingredient Listed
Carbohydrates/sugars	Black beans, chickpeas, rice, spelt, corn, quinoa, hydrated bulgur, breadcrumbs, wheat and lupin flour, texturized rice, sugar cane.
Fat/Saturated fat	Sunflower oil, high oleic sunflower oil, turnip oil, coconut oil, olive oil, extra-virgin olive oil, sunflower seed oil, rapeseed oil, variable proportion of vegetable oils.
Protein	Soy protein, rehydrated soy protein, pea protein, pea protein isolate, soybeans, hydrolyzed vegetable protein, wheat gluten, rehydrated wheat protein, rice protein, egg white.
Dietary Fiber	Vegetable fiber, bamboo fiber, pea fiber, wheat fiber, plantain fiber, soy fiber.
Animal protein as an ingredient	Whole eggs (egg white and yolk) or powdered eggs, dairy products (milk, cheese).
Vitamins and Minerals	Iron was added in 11/148 products, vit B12 was added in 10/148 products.

**Table 5 nutrients-15-01325-t005:** Plant-based meat alternatives categories with an organic ingredient.

Plant-Based Products (*n* = 148)				
Category	Total	Organic Vegetable Protein	Organic Animal Protein	Organic Fat
Plant-based sausages	21	11	2	11
Plant-based nuggets and breaded food	25	3	2	3
Plant-based meatballs	9	2	2	2
Vegetarian cold cuts	17	3	0	3
Veggie patties	29	18	1	18
“Beyond Meat-type” hamburger patties	20	5	0	5
“Chicken-type” strips	10	3	0	0
Plant-based mince	17	5	0	3

**Table 6 nutrients-15-01325-t006:** Nutrients/100 g (median, minimum–maximum) and NOVA classification.

	Nutrient Criteria								
Plant-Based Meat Alternatives Categories	Energy (kcal)	Protein (g)	Fat (g)	Saturated Fat (g)	Carbohydrate (g)	Sugars (g)	Dietary Fiber (g)	Salt (g)	NOVA
	Median (Min–Max)	Median (Min–Max)	Median (Min–Max)	Median (Min–Max)	Median (Min–Max)	Median (Min–Max)	Median (Min–Max)	Median (Min–Max)	Median (Min–Max)
Plant-based sausages	254 (153–312)	17.0 (8.00–27.0)	15.0 (6.00–25.1)	1.90 (0.20–18.6)	6.80 (0.20–17.0)	1.20 (0.30–3.40)	3.00 (0.50–6.60)	1.50 (0.20–2.50)	4
Plant-based nuggets and breaded foods	229 (126–295)	12.3 (3.60–20.0)	10.2 (3.90–17.4)	1.20 (0.70–3.80)	17.9 (12.8–37.0)	1.90 (0.00–9.50)	3.80 (1.60–6.00)	1.23 (0.60–2.10)	4
Plant-based meatballs	189 (143–267)	16.0 (7.40–22.0)	9.60 (4.40–15.0)	1.10 (0.60–1.80)	8.30 (2.00–19.0)	2.00 (0.90–4.10)	5.20 (1.50–9.00)	1.42 (0.90–1.60)	4
Vegetarian cold cuts	174 (119–254)	7.30 (4.00–35.3)	13.0 (1.10–16.0)	1.60 (1.00–6.30)	5.10 (1.00–10.0)	1.00 (0.49–3.80)	3.30 (0.50–5.00)	2.00 (1.10–2.60)	4
Veggie patties	202 (144–288)	14.0 (4.30–23.0)	9.50 (5.00–19.0)	1.20 (0.50–2.20)	15.6 (4.50–33.5)	2.70 (0.10–6.80)	3.85 (2.30–7.80)	1.19 (0.48–1.80)	4
“Beyond Meat-type” hamburger patties	201 (125–290)	15.5 (8.90–22.0)	9.25 (4.50–19.0)	1.25 (0.00–9.00)	8.90 (1.90–22.2)	1.25 (0.00–4.00)	4.00 (1.30–7.70)	1.35 (0.00–2.50)	4
“Chicken-type” strips	154 (129–226)	19.0 (15.5–24.5)	5.30 (3.00–11.0)	0.55 (0.40–1.10)	2.65 (1.20–10.0)	0.65 (0.00–2.10)	6.40 (4.80–9.80)	1.40 (1.00–2.50)	4
Plant-based mince	320 (142–400)	47.30 (6.60–55.7)	8.80 (0.50–17.0)	1.50 (0.10–8.70)	13.0 (3.60–34.79)	5.80 (0.00–13.0)	6.00 (2.50–17.0)	0.07 (0.00–2.00)	3 and 4

## Data Availability

The data presented in this study are available on request from the corresponding author.

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
