# Peer review of "Nutritional Assessment of Plant-Based Meat Alternatives: A Comparison of Nutritional Information of Plant-Based Meat Alternatives in Spanish Supermarkets"

_nutrients, 2023, doi:10.3390/nu15061325_

Round 1
Reviewer 1 Report
This manuscript investigates the nutritional profiles of plant-based meat alternatives in Spain. While the paper adds an interesting local aspect for the area of plant-based meat alternatives, the reviewer would suggest discussing the results in a broader context (comparison to meat, comparison to other countries, discuss quantity but also quality of protein, dietary fiber, micronutrients, interactions with anti-nutritive factors). In particular the discussion should be substantially reworked to include these aspects. Furthermore, the discussion should be strengthened with strengths, limitations and further research needs or other requied aspect relevant for consumers, food developers, policy makers. Following, the conclusion should be reworked based on this adaptation.
Abstract overall: I suggest adding some nutrient ranges in this sentence Of the 148 products, the majority were low in sugars but moderate in carbohydrates, total and saturated fat, and high in salt.
Line 12: IARC - do not use the abbreviation without introducing it
Line 25: what is meant with a variable nutritional composition? Is it heterogenous within and/ or between categories?
Line 40: increased by 26% (to add)
Line 48 and 49: There are more and more studies on the nutritional assessment of plant-based meat analogues, see for example https://www.tandfonline.com/doi/full/10.1080/09637486.2022.2078286 with several references. Therefore, rephrase the sentence
Line 66 – 69 – sentence is difficult to understand, consider rephrasing
Table 1: A lot of text, could it be described in a simpler way?
e.g. < 40 kcal/ 100 g solids
< 20 kcal/ 100 g liquids
Instead of use conditions requirement for claim
Statistics: was it tested whether data are normally distributed? Otherwise rather use median with a range min – max, that is more meaningful
Table 5 add n= 148
Table 6: Dietary Fiber Plant-based sausages 3.14 ± 34.4 ? is this correct? Same issue for Plant-based nuggets and meatballs
Discussion – the discussion should be reworked to compare with the corresponding meat, discuss within and between category differences and compare all results in the international context. In addition, add strength, limitations and what is required in this context (better guidance to consumers or another aspect..).
Line 196 – how representative is the study for Spain? Give more indications
Line 204, 206 – Author without initials of first name
Line 210 – only energy content is compared with other studies. Discuss the data and compare also the other results in the international context.
Line 224 – 248 – nothing is said about the protein quality and digestibility. This is an important factor that needs to be discussed.
Line 261 – 263 – the sentence is placed there a bit out of the context. Also – meaning unclear. Does it mean nutrition profiles for the development of PBMAs or nutrition guidelines for consumers?
Line 273 - The procedure suggested for these health connections – check more suitable wording in English
Conclusion – rework the conclusion based on the additions in the discussion
Line 282 – 283 – great content of protein and fiber but what about quality, digestibility
Line 285 – 288 – for the conclusion that PBMAs are a healthier option than meat you need to compare with meat – this is lacking. The whole nutritional profile including antinutrients, vitamins, minerals, availability and digestibility needs to be discussed. Furthermore, what is meant with regulation is needed – guidelines to consumers of another labelling or another aspect?
Line 287 – 287 what is meant by Complete and evidence-based information may be required to advertise these types of products in the context of a healthy diet? Be very concrete what is meant here. Here, limitations of your study might be repeated.
Author Response
"Please see the attachment."

Reviewer 2 Report
The article entitled "Nutritional evaluation of vegetable alternatives to meat. A comparison of the nutritional information of vegetable alternatives to meat in Spanish supermarkets" is very interesting. The aims to the autors was to evaluate the nutritional composition and processing degree and to perform a quantitative analysis of the ingredients of PBMA products currently available in Spanish supermarkets.
I would like to ask the authors questions on how the nutritional data of foods were collected: who went to the supermarkets? How were the nutritional values of foods reported? (for example, were they copied by the operator or were they photographed?).
I suggest the authors to describe these aspects in the methods paragraph.
Author Response
"Please see the attachment."

Round 2
Reviewer 1 Report
Thank you for responding to my comments and reworking the manuscript. From my point of view I would accept it now for publication.